# Effectiveness of integrated care for older adults with depression and hypertension in rural China: A cluster randomized controlled trial

Shulin Chen[1]☯, Yeates Conwell[2]☯*, Jiang Xue[1], Lydia Li[3], Tingjie Zhao[4], Wan Tang[4], Hillary Bogner[5], Hengjin Dong[6]

1 Department of Psychology, Zhejiang University, Hangzhou, China, 2 Department of Psychiatry, University of Rochester, Rochester, New York, United States of America, 3 School of Social Work, University of Michigan, Ann Arbor, Michigan, United States of America, 4 Department of Biostatistics and Data Science, Tulane University, New Orleans, Louisiana, United States of America, 5 Department of Family Medicine, University of Pennsylvania, Philadelphia, Philadelphia, United States of America, 6 Center for Health Policy Studies, School of Public Health, Zhejiang University, Hangzhou, China

☯ These authors contributed equally to this work.
* Yeates_Conwell@urmc.rochester.edu

**Data Availability Statement:** All data files and code book are published and available from the Harvard Dataverse – https://doi.org/10.7910/DVN/6RWH41.

## Abstract

### Background

Effectiveness of integrated care management for common, comorbid physical and mental disorders has been insufficiently examined in low- and middle-income countries (LMICs). We tested hypotheses that older adults treated in rural Chinese primary care clinics with integrated care management of comorbid depression and hypertension (HTN) would show greater improvements in depression symptom severity and HTN control than those who received usual care.

### Methods and findings

The study, registered with ClinicalTrials.gov as Identifier NCT01938963, was a cluster randomized controlled trial with 12-month follow-up conducted from January 1, 2014 through September 30, 2018, with analyses conducted in 2020 to 2021. Participants were residents of 218 rural villages located in 10 randomly selected townships of Zhejiang Province, China. Each village hosts 1 primary care clinic that serves all residents. Ten townships, each containing approximately 20 villages, were randomly selected to deliver either the Chinese Older Adult Collaborations in Health (COACH) intervention or enhanced care-as-usual (eCAU) to eligible village clinic patients. The COACH intervention consisted of algorithm-driven treatment of depression and HTN by village primary care doctors supported by village lay workers with telephone consultation from centrally located psychiatrists. Participants included clinic patients aged ≥60 years with a diagnosis of HTN and clinically significant depressive symptoms (Patient Health Questionnaire-9 [PHQ-9] score ≥10). Of 2,899

**Funding:** The COACH Study was supported by a grant from the National Institute of Mental Health <https://www.nimh.nih.gov> of the U.S. National Institutes of Health under Award Number R01MH100298 to YC and CS as multiple principal investigators. The funders had no role in study design, data collection and analysis, decision to publish, or preparation of the manuscript.

**Competing interests:** The authors have declared that no competing interests exist.

**Abbreviations:** ADL, activity of daily living; AW, aging worker; BP, blood pressure; CDC, China Center for Disease Control and Prevention; COACH, Chinese Older Adult Collaborations in Health; eCAU, enhanced care-as-usual; GLMM, generalized linear mixed effect model; HDRS, Hamilton Depression Rating Scale; HTN, hypertension; IADL, instrumental activity of daily living; ICC, intraclass correlation; LMIC, low- and middle-income country; LSNS, Lubben Social Network Scale; MINI, Mini-International Neuropsychiatric Interview; MOS-SSS-C, Chinese version of the Medical Outcomes Study Social Support Survey; OR, odds ratio; PCP, primary care provider; PHQ-9, Patient Health Questionnaire-9; RA, research assistant; RCT, randomized control trial; SIS, Six-Item Screener; STAGED, Duke Somatic Treatment Algorithm for Geriatric Depression; WGEE, weighted generalized estimating equation.

eligible village residents, 2,365 (82%) agreed to participate. They had a mean age of 74.5 years, 67% were women, 55% had no schooling, 59% were married, and 20% lived alone.

Observers, older adult participants, and their primary care providers (PCPs) were blinded to study hypotheses but not to group assignment. Primary outcomes were change in depression symptom severity as measured by the Hamilton Depression Rating Scale (HDRS) total score and the proportion with controlled HTN, defined as systolic blood pressure (BP) <130 mm Hg or diastolic BP <80 for participants with diabetes mellitus, coronary heart disease, or renal disease, and systolic BP <140 or diastolic BP <90 for all others. Analyses were conducted using generalized linear mixed effect models with intention to treat.

Sixty-seven of 1,133 participants assigned to eCAU and 85 of 1,232 COACH participants were lost to follow-up over 12 months. Thirty-six participants died of natural causes, 22 in the COACH arm and 14 receiving eCAU. Forty COACH participants discontinued antidepressant medication due to side effects. Compared with participants who received eCAU, COACH participants showed greater reduction in depressive symptoms (Cohen's d [±SD] = −1.43 [−1.71, −1.15]; $p < 0.001$) and greater likelihood of achieving HTN control (odds ratio [OR] [95% CI] = 18.24 [8.40, 39.63]; $p < 0.001$).

Limitations of the study include the inability to mask research assessors and participants to which condition a village was assigned, and lack of information about participants' adherence to recommendations for lifestyle and medication management of HTN and depression. Generalizability of the model to other regions of China or other LMICs may be limited.

## Conclusions

The COACH model of integrated care management resulted in greater improvement in both depression symptom severity and HTN control among older adult residents of rural Chinese villages who had both conditions than did eCAU.

## Trial registration

ClinicalTrials.gov Identifier NCT01938963 https://clinicaltrials.gov/ct2/show/NCT01938963.

Author summary

**Why was this study done?**

- The combination of depression and hypertension (HTN) is common in later life and makes both conditions worse.

- Studies in high-income countries have found that older adults with both medical illness and clinically significant depression have better outcomes when the care delivered by their primary care doctor is integrated with mental healthcare.

- The integrated approach to care for older adults has been insufficiently tested in low- and middle-income countries (LMICs).

**What did the researchers do and find?**

- We developed a model of integrated care for combined depression and HTN among older adults in rural village primary care clinics in China and compared it with usual approaches to care.

- The intervention, called Chinese Older Adult Collaborations in Health (COACH), was administered by a team consisting of the village clinic's primary care doctor, a village resident called an "Aging Worker" to help reinforce treatment and address social factors affecting health, and telephone-based consultation with a psychiatrist.

- Over 12 months of participation, the 1,232 participants who received the COACH intervention had a significantly greater improvement in both depressive symptoms and the rate of HTN control than the 1,133 participants who received usual care.

**What do these findings mean?**

- Integrated care management appears effective for management of comorbid depression and HTN in Chinese older adults in underresourced primary care settings that have limited access to mental healthcare.

- The unique benefits to depression treatment and hypertension control of antidepressant prescription and Aging Worker support, and the addition of easily administered psychotherapy to the model, require further study.

- Wide dissemination of the model in areas with few mental health resources will likely require use of telehealth technologies to enable better access to care.

## Introduction

Older adults are the fastest growing segment of the world's population, a pattern especially pronounced in low- and middle-income countries (LMICs). China has an estimated 254 million people over the age of 60 years [1]. Depression and hypertension (HTN) are among the most common disorders of later life [2,3]. Known as "the silent killer" [4], HTN is a major cause of strokes and ischemic heart disease, while depressive disorders are the third leading cause of years lived with disability worldwide [5]. Depression and HTN commonly coexist, making their management more complex and clinical outcomes worse [6–11].

Primary care providers (PCPs) in rural China receive little training in the detection, diagnosis, and management of common mental disorders, and mental health specialty care is difficult to access [6,12]. Consequently, late life affective disorders are rarely treated. In a naturalistic longitudinal study of older urban residents in China, Chen and colleagues found that only 1% of those with a major depressive episode received treatment over 12 months [13].

In high-income countries, collaborative depression care management models [14–16] have proven effective in managing depression and comorbid medical conditions while also reducing costs in primary care settings [17–19]. The collaborative care approach brings together PCPs, care managers, and psychiatric consultants to deliver care and monitor patient progress. The team employs systematic screening, evidence-based interventions administered with

treatment algorithms, and decision support tools. Although well tested and widely disseminated in high-income regions, few studies of the model have been conducted in LMICs that used a randomized control trial (RCT) design, had a focus on older adults, or examined comorbid depression and medical illness [20–22].

We compared a collaborative depression care management intervention called Chinese Older Adult Collaborations in Health (COACH) to enhanced care-as-usual (eCAU) for the treatment of comorbid depression and HTN in older residents of rural Chinese village clinics. Our primary hypotheses were that older adults with comorbid depression and HTN who received the COACH intervention would show greater improvements in both depressive symptom severity and HTN control than those who received eCAU over 12 months. We also explored post hoc whether acceptance of antidepressant medications by those in the COACH intervention was associated with treatment response.

## Methods

### Study design

The COACH Study is a cluster RCT with 12-month follow-up conducted in Tonglu and Jiande counties of Zhejiang Province, China. Together, these 2 counties have a total of 920,000 residents distributed across 30 townships with a total of 602 villages. The health needs of each village are provided by a clinic staffed by 1 PCP without other nursing support. Mental healthcare is provided by 1 county-level mental hospital, and residents' social needs are provided by the village's Aging Association staffed by local residents.

### Randomization and masking

The original design planned for village to be the unit of randomization [23]. However, it became clear prior to initiation of the study that the chance of contamination bias between study conditions was substantial due to interactions between neighboring village PCPs. Therefore, randomization was based on the township where the villages were located, while outcomes pertain to the individual participant. Ten townships located in Tonglu and Jiande Counties were randomly selected by a computer algorithm administered by the study statistician to assure that no two shared a common boundary. Each township contained from 18 to 25 villages, and each village contained 1 primary care clinic. Five townships were selected for each arm of the study, with all villages and their associated clinics in each township being assigned either to deliver the COACH intervention or eCAU to eligible patients. The PCP and aging workers (AWs) for COACH intervention clinics (see "Intervention-COACH" below) were approached by the research team for agreement to participate. One village assigned to the COACH arm refused. The final numbers of villages in the COACH and eCAU intervention arms were 102 and 116, respectively.

Masking was not feasible for research participants or assessors. All were blinded, however, to the study objectives and hypotheses.

### Participants

Potential participants were initially identified by review of village clinic electronic medical records by the PCP. Inclusion criteria were registration in the village's primary care clinic; age ≥60 years; and a chart diagnosis of HTN. All clinic patients in rural China with a diagnosis of HTN are prescribed antihypertensive medications consistent with the directives of the China Center for Disease Control and Prevention (CDC). PCPs trained in use of the Patient Health Questionnaire-9 (PHQ-9) [24] administered the measure to each potential participant. Those

recording a total score of ≥10, indicative of clinically significant depression, were eligible to meet with the study personnel who further assessed for intact cognitive functioning (Six-Item Screener [SIS] score <3) [25] and willingness to give written informed consent. Exclusion criteria included mania, psychosis, or alcohol abuse or dependence active in past 6 months based on the Mini-International Neuropsychiatric Interview (MINI) [26] administered by research assessors at baseline; and acute suicide risk determined by the potential participant's PCP. Participants in the COACH group could decline antidepressant prescription and remain in the study. Potentially eligible participants were invited to meet in their homes or the clinic with a research assistant (RA) who introduced the study, assessed eligibility, and obtained written informed consent.

## Assessment

Trained research assessors administered study measures in the participant's home or the clinic at study entry and 3, 6, 9, and 12 months later. Sociodemographic information collected at baseline included participants' age, sex, marital status and living situation (alone, with spouse and/or children, or with others), years of education, employment status, religion (Buddhist, Christian, Islam, none), and level of economic satisfaction (very sufficient to very poor). Other measures assessed health-related quality of life using the WHOQOL-BREF [27,28], the number of medical comorbidities recorded in the participant's medical record, impairment in basic (ADL) and instrumental activities of daily living (IADLs) [29], and social support using the Chinese version of the Medical Outcomes Study Social Support Survey (MOS-SSS-C) [30]. Social network size was assessed with the Lubben Social Network Scale (LSNS), a continuous measure of perceived social support received from family and friends [31].

The primary outcome measure of depressive symptoms was total score on a valid and reliable Chinese translation of the Hamilton Depression Rating Scale (HDRS) [32]. The 17-item HDRS assesses symptoms of depressive illness including mood, anxiety, neurovegetative symptoms of energy, appetite and sleep, anhedonia, psychomotor change, and somatic concerns. Each item is scored on a 3- or 5-point scale with total score representing their sum (range, 0 to 52). Higher scores indicate greater depressive symptom severity.

The primary outcome of HTN control was based on BP measured at baseline and each follow-up point according to China CDC standards [33]. The primary HTN outcome was uncontrolled HTN, defined as systolic blood pressure (BP) ≥130 mm Hg or diastolic BP ≥80 for patients with diabetes mellitus, coronary heart disease, or renal disease, and systolic BP ≥140 or diastolic BP ≥90 for all others.

## Comparison condition—Enhanced Care-as-Usual (eCAU)

PCPs in village clinics ordinarily have 3 years of medical education after high school [34]. They receive periodic in-service education that includes HTN management using CDC practice guidelines that address detection, management by both pharmacologic and nonpharmacologic means, and guidance on when to make referral to specialized care [35]. No training or guidelines are provided to PCPs in the diagnosis and treatment of affective disorders. Rather, when PCPs suspect mental illness in their patients, they ordinarily recommend the patient go to the County Mental Hospital. PCPs cannot initiate antidepressant treatment but can renew prescriptions initiated by County Mental Hospital psychiatrists.

We refer to care as CAU as "enhanced" (eCAU) because PCPs were told when their patients screened positive for depression and were provided with copies of antidepressant treatment guidelines adapted from the Duke Somatic Treatment Algorithm for Geriatric Depression (STAGED) [36].

## Intervention—COACH

The COACH intervention team in each village consisted of the clinic's PCP, the village AW, and a county hospital psychiatrist. The background and qualifications of the PCP are as described above for eCAU. Each village, in addition to 1 PCP, has 1 AW employed part-time by the village leaders. The great majority are women with a middle-school education who receive in-service training from the Bureau of Civil Affairs in addressing the villagers' social needs. The COACH intervention linked the PCP and AW of each village remotely by telephone with 1 psychiatrist based at the County Mental Hospital. There was no turnover in PCPs, AWs, or psychiatrists during any village's 12-month participation in the study and, because the staffing of health and social services does not differ between villages in the counties involved, there was no need for tailoring of the intervention between them.

Recruitment of villages and participants within them took place over 4 waves. Training for each wave took place over 4 days immediately preceding the start of the intervention for that cohort. It included separate groups for instruction in the roles of the PCP, AW, and psychiatrist and joint training in how to work together in delivering the intervention collaboratively and with fidelity.

COACH PCPs received training in depression assessment, case management using the toolkit adapted from the MacArthur Initiative on Depression in Primary Care [37], and use of the STAGED antidepressant treatment guidelines [36]. There were no restrictions on concomitant care or treatment. The expectation was for PCPs to see participants at baseline and again at regular intervals, at least monthly, for follow-up depression screening (PHQ-9) and BP checks.

The AWs' role was to educate participants about their conditions followed by goal setting and collaborative development of behavior change strategies. Their training curriculum included an overview of depression, HTN, and their relationship; principles of disease self-management; psychosocial assessment and care planning; psychoeducation with older adults and their families; and ethical standards including confidentiality. They did not receive training in manualized psychosocial interventions. During monthly home visits, AWs helped the older person to set behavioral goals, review progress, and modify the plan if needed. In addition, the AWs created social opportunities for older persons, for example, by organizing social activities and educational workshops on a monthly basis.

The psychiatrist provided initial in-person consultation with the PCP in the village clinic and made the initial prescription of antidepressants as indicated. Thereafter, they met with the PCP and AW on a monthly basis by telephone to review cases and make recommendations.

Training materials are available in Chinese on request from the first author. See Supporting information (S1 Table) for listing of roles and responsibilities of each COACH team member —PCP, AW, and psychiatrist.

## Intervention procedures and outcomes

At baseline, the AW assessed the person's social supports, lifestyle, and functional, nutritional, and financial status in the participant's home. The AW, PCP, and psychiatrist reviewed all the assessment data and constructed a care plan.

The PCP and AW met with each participant monthly in the home or clinic to monitor BP and depressive symptoms, assess antidepressant and antihypertensive use and side effects, and provide support to the patient and family. The PCP and AW met weekly in person to review their shared caseload, and with the psychiatrist monthly by telephone to assess progress and update the treatment plan as indicated.

## Primary outcomes

Primary outcomes of the study were change in depression symptom severity as measured by the HDRS at baseline and follow-up assessments, and the proportions (%) of participants whose HTN was controlled. Initial plans to measure adherence to medications could not be completed for administrative reasons.

## Statistical analysis

The study aimed to recruit 1,200 participants to each study arm. Because the intraclass correlation (ICC) at the town level was likely to be low, the study used a 3-level nested design in which the power was based on the ICC among the patients within the village and serial correlation between repeated assessments within the patient. With this number of participants, setting the serial correlation at 0.5 and varying the ICC over 0.05, 0.1, and 0.2, and assuming a two-sided type I error = 0.05, power = 0.8, and an attrition rate of 20%, the detectable effect size ranged from 0.17 to 0.26 for the continuous outcome of depression symptom change. A 3-point between-group difference, considered the minimal clinically important difference in HDRS score, corresponds to an effect size of 0.5. Therefore, our study was amply powered for that outcome [38]. For HTN control, we also set the serial correlation at 0.5 and varied the percent of variance between patients within the village clinic over 0.05, 0.1, and 0.2. Based on two-sided type I error = 0.05, power = 0.8, base rate 0.5 (most conservative), and 20% attrition rate, the detectable between-group proportion with 2,400 participants ranged from 11% to 17%, well within the range of clinically meaningful differences in primary care settings.

We compared baseline characteristics between groups using $t$ test and chi-squared test. Given that the randomization was at the township level and the number of clusters created was relatively small, we treated characteristics that significantly differentiated the 2 groups at $p < 0.05$ as covariates when testing between-group differences using longitudinal models.

To test our hypotheses that COACH participants would show greater improvements in HTN control and depressive symptom severity as measured by the HDRS than eCAU participants over 12 months of involvement in the study, we modeled the repeatedly assessed variables of depressive symptom severity and of HTN control using generalized linear mixed effect models (GLMMs) and weighted generalized estimating equations (WGEEs). Since the GLMM and WGEE were consistent, only the GLMM results were reported [39]. Analyses were performed as intention-to-treat using all participants recruited from the 10 towns being randomized. Random effects of towns, villages, and participants were used to account for the within-cluster associations within the towns and villages, as well as repeated measurements within the participants. For each outcome, time and intervention were predictors adjusting for covariates. We assessed the potential interaction between time and intervention using linear contrasts to assess COACH versus eCAU differences over the 12-month period as well as any subintervals within this period.

With regard to model assumptions, the functional forms or scales of the quantitative covariates were examined using the fractional polynomial approach of Royston and Altman [40], and the final models were selected based on the Akaike information criterion. We further calculated the R squared, both marginal and conditional, as proposed by Vonesh and colleagues [41] to assess the goodness of fit of these selected GLMMs. The marginal R squared ranged from 0.24 to 0.29 and the conditional R squared ranged from 0.65 to 0.83, all of which fall in the acceptable range for psychosocial studies. In order to check the normality of residuals, we used QQ-plot to demonstrate that all models were normally distributed with constant variance.

Following completion of analyses planned a priori, we conducted a post hoc exploratory comparison of participants within the COACH group who had accepted prescription of an

antidepressant as a component of their treatment (Antidep[+]) with those who had not (Antidep[−]). The purpose was to explore whether antidepressant use may explain observed differences in depression and HTN treatment response between the 2 subgroups.

Statistical analyses were conducted with Statistical Analysis System software version 9.4 or above. The study protocol was prospectively registered with Clinicaltrials.gov as Identifier NCT01938963 [42].

### Ethical procedures

Written informed consent was obtained from all participants prior to their enrollment and baseline study assessment. No compensation was provided for their participation. The study was reviewed and approved by the institutional review boards of Zhejiang University, the University of Rochester, and the US National Institute of Mental Health (NIMH), and a Data Safety Monitoring Board maintained oversight of its implementation.

The study is reported as per the Consolidated Standards of Reporting Trials (CONSORT) Extension to Cluster Randomised Trials [43] (see Supporting information S2 Appendix) and the Template for Intervention Description and Replication (TIDieR) checklist (S3 Appendix).

## Results

There were 2,365 participants enrolled from 218 villages between January 1, 2014 and September 30, 2017, yielding 1,232 participants in COACH intervention villages and 1,133 in eCAU. All village clinics remained actively engaged in the trial throughout 12 months of participation with 85 and 67 participants lost to follow-up in the COACH and eCAU groups, respectively. Reasons for lost to follow-up, categorized as deceased, unable to contact, and refused further assessment, are depicted for COACH and eCAU groups in Fig 1, the study's CONSORT diagram. Table 1 provides the demographic and baseline characteristics of the samples. Because of baseline differences between groups at the $p < 0.05$ level, religion, employment status, all WHOQOL-BREF scales, MOS-SSS-C, Social Network Size, count of comorbidities, and HTN control were included as covariates in subsequent analyses.

COACH team adherence to the intervention was tracked by activity logs maintained by each team member (see Supporting information S3 Table). PCPs and AWs were expected to meet with COACH participants monthly. On average, they met with the participants under their care an average of 11.6 and 8.3 times, respectively, over the 12 months of study engagement. AWs organized an average of 9.8 community activities that participants were encouraged to attend, of 12 expected over 12 months. The PCP and AW in each COACH clinic met together an average of 56.6 times during the year to coordinate care of their mutual patients, and of 12 anticipated meetings of the team by telephone with the consulting psychiatrist, they met an average of 7.9 times.

Results of analyses unadjusted for covariates are available in S6 Table. In Table 2, which depicts results of analyses adjusted for covariates, HDRS total scores [SD] decreased steadily over 12 months of study participation for those receiving COACH, albeit to levels still considered to be moderately symptomatic (HDRS [SD] = 12.7 [4.2]). eCAU participants had a smaller reduction in HDRS score over 12 months from 21.8 [3.6] to 18.8 [4.7]. The group × time interaction was significant with a large effect size, suggesting that the COACH group had a faster reduction in depressive symptom severity than the eCAU group (F = 217.38; $p < 0.001$). The ICCs at the patient (repeated outcomes of the same subject) and village (individuals within the same village) levels were 0.54 and 0.08, respectively.

HTN control was more likely to be achieved by participants in COACH villages, increasing from 25.1% to 71.6% over the 12 months of study participation, than those who received

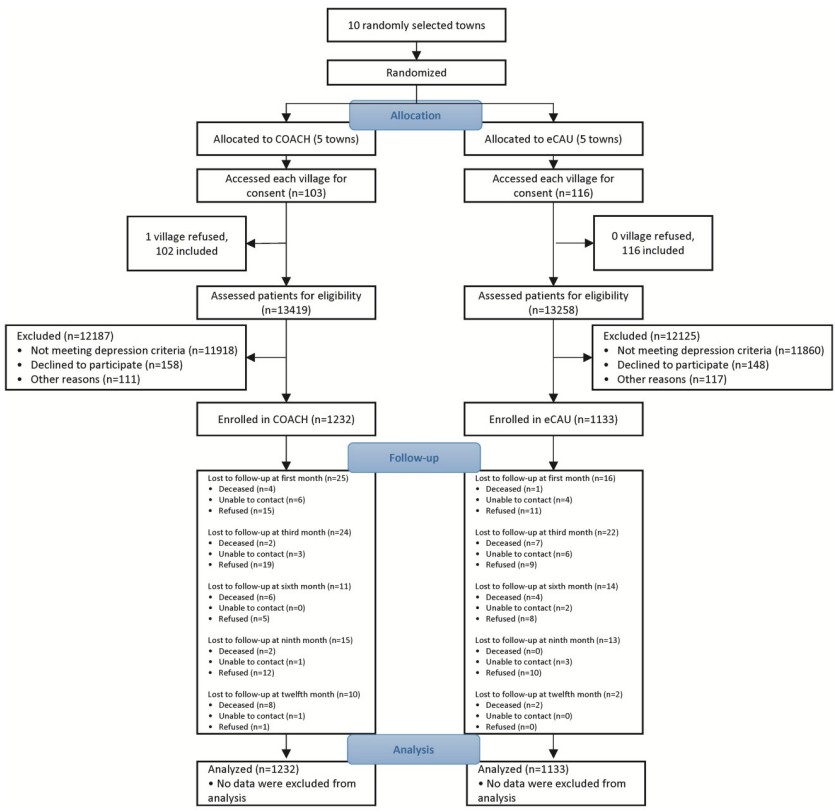

**Fig 1. CONSORT diagram of COACH Study participant flow.**

eCAU (from 20.2% to 40.9%). The group × time interaction was significant with a large effect size for HTN control as well (F = 12.74; $p < 0.001$). The ICCs at the patient and village levels were 0.63 and 0.20, respectively. On completing the study, COACH participants showed greater reduction in depressive symptoms (Cohen's d [SD] = −1.43 [−1.71, −1.15]) and greater likelihood of achieving HTN control (odds ratio [OR] [95% CI] = 18.24 [8.40, 39.63]).

There were no unintended or reportable serious adverse events.

A substantial minority (*n* = 518; 42%) of COACH group participants declined to take antidepressant medications as part of their treatment. Comparing those who did accept antidepressants (Antidep[+]) with those who did not (Antidep[−]), there were no notable differences in demographic and clinical characteristics at baseline. Only HDRS score distinguished the 2 groups (Antidep[+] [SD] = 22.6 (4.8); Antidep[−] [SD] = 21.4 [4.0]; $p < 0.001$), although the absolute difference of 1.2 HDRS points is unlikely to be clinically meaningful. Figs 2 and 3 allow for visual comparison of eCAU with Antidep[+] and Antidep[−] subgroups. The COACH Antidep[−] group had a pattern of response in HDRS total score between that of eCAU and Antidep[+] (Fig 2).

With regard to HTN control, a different pattern emerged. There was no meaningful difference between the COACH subgroups in the proportions that achieved HTN control (Antidep [+] = 27% versus Antidep[−] = 23%; $p = 0.11$), but improvements in HTN control were clearly greater for both COACH subgroups than for eCAU participants as depicted in Fig 3. The strong impact of the intervention appears equivalent with regard to HTN for both COACH subgroups relative to eCAU.

**Table 1. Demographic and clinical characteristics of participants at baseline (N = 2,365).**

| Characteristic | Group | | | *p*-value |
|---|---|---|---|---|
| | TOTAL | eCAU | COACH | |
| Sample size, n (%) | 2,365 | 1,133/2,365 (48) | 1,232/2,365 (52) | |
| Age, mean (SD) | 74.46 (8.23) | 74.58 (8.34) | 74.35 (8.13) | 0.52 |
| 60–69 years | 783 (33) | 378 (33) | 405 (33) | |
| 70–79 | 849 (36) | 396 (35) | 453 (37) | |
| 80–89 | 665 (28) | 318 (28) | 347 (28) | |
| ≥90 | 68 (3) | 41 (4) | 27 (2) | |
| Sex, n (%) | | | | 0.79 |
| Male | 789/2,365 (33) | 381/1,133 (34) | 408/1,232 (33) | |
| Female | 1,576 (67) | 752 (66) | 824 (67) | |
| Marriage status, n (%) | | | | 0.62 |
| Married | 1,389/2,365 (59) | 675/1,133 (60) | 723/1,232 (59) | |
| Divorce | 15 (1) | 6 (1) | 9 (1) | |
| Never Married | 25 (1) | 15 (1) | 10 (1) | |
| Widowed | 925 (39) | 437 (38) | 488 (39) | |
| Separation | 2 (0) | 0 (0) | 2 (0) | |
| Living situation, n (%) | | | | 0.83 |
| Alone | 482/2,365 (20) | 235/1,133 (21) | 247/1,232 (20) | |
| With Spouse and/or Children | 1,843 (78) | 877 (77) | 966 (78) | |
| With Others | 40 (2) | 21 (2) | 19 (2) | |
| Education, n (%) | | | | 0.86 |
| No Schooling | 1,297/2,365 (55) | 621/1,133 (55) | 676/1,232 (55) | |
| 1–5 Years | 926 (39) | 447 (39) | 479 (39) | |
| ≥6 Years | 142 (6) | 65 (6) | 77 (6) | |
| Employment status, n (%) | | | | <0.001 |
| Full-time | 381/2,365 (16) | 222/1,133 (20) | 159/1,232 (13) | |
| Part-time | 328 (14) | 170 (15) | 158 (13) | |
| Not Employed | 1,656 (70) | 741 (65) | 915 (74) | |
| Religion, n (%) | | | | 0.003 |
| Buddhist | 352/2,365 (15) | 186/1,133 (16) | 166/1,232 (14) | |
| Christian | 106 (4) | 65 (6) | 41 (3) | |
| Islam | 1 (0) | 1 (0) | 0 (0) | |
| No religion | 1,906 (81) | 881 (78) | 1,025 (83) | |
| Economic satisfaction, n (%) | | | | 0.012 |
| Very sufficient | 31/2,365 (1) | 14/1,133 (1) | 17/1,232 (1) | |
| Sufficient | 370 (16) | 185 (16) | 185 (15) | |
| Moderate | 1,463 (62) | 669 (59) | 794 (64) | |
| Poor | 451 (19) | 233 (21) | 218 (18) | |
| Very Poor | 50 (2) | 32 (3) | 18 (2) | |
| Count of comorbidities, mean (SD) | 1.65 (1.45) | 1.57 (1.41) | 1.72 (1.48) | 0.012 |
| HDRS score, mean (SD) | 21.92 (4.10) | 21.76 (3.58) | 22.07 (4.52) | 0.068 |
| WHOQOL-BREF, mean (SD) | | | | |
| Overall quality of life | 2.90 (0.77) | 2.86 (0.77) | 2.91 (0.77) | 0.52 |
| Overall health condition | 2.66 (0.76) | 2.57 (0.76) | 2.75 (0.75) | <0.001 |
| Physical well-being | 18.96 (2.86) | 18.08 (2.46) | 19.79 (2.95) | <0.001 |
| Psychological well-being | 16.24 (2.90) | 15.68 (2.66) | 16.76 (3.02) | <0.001 |
| Social well-being | 8.88 (2.31) | 8.70 (1.74) | 9.04 (2.72) | 0.003 |

(*Continued*)

**Table 1.** (Continued)

| Characteristic | Group | | | p-value |
|---|---|---|---|---|
| | TOTAL | eCAU | COACH | |
| Environment | 24.96 (4.91) | 23.94 (4.04) | 25.90 (5.44) | <0.001 |
| ADL, mean (SD) | 21.97 (10.16) | 22.24 (10.20) | 21.72 (10.13) | 0.22 |
| IADL, mean (SD) | 5.57 (2.36) | 5.53 (2.27) | 5.61 (2.44) | 0.41 |
| MOS-SSS-C, mean (SD) | 2.83 (0.86) | 2.93 (0.78) | 2.73 (0.93) | <0.001 |
| Social Network Size, mean (SD) | 24.93 (10.47) | 26.64 (11.28) | 23.48 (9.40) | <0.001 |
| HTN control, n (%) | | | | 0.005 |
| Controlled | 538/2,365 (23) | 229/1,133 (20) | 309/1,232 (25) | |
| Uncontrolled | 1,827 (77) | 904 (80) | 923 (75) | |
| Agreed to take antidepressants | n/a | n/a | 714 (58) | |

ADL, Ability of Daily Living Scale; COACH, Chinese Older Adults Collaborations in Health; eCAU, enhanced Care-as-Usual; HDRS, Hamilton Depression Rating Scale; HTN, hypertension; IADL, Instrumental Activity of Daily Living Scale; MOS-SSS-C, the Chinese version of the Medical Outcomes Study Social Support Survey; SD, standard deviation; WHOQOL-BREF, the World Health Organization Quality of Life Questionnaire abbreviated version.

The data for Figs 2 and 3 are provided as Supporting information (S4 and S5 Tables, respectively).

## Discussion

Consistent with our hypotheses, the study found that older rural village clinic patients in China who received the COACH intervention had significantly greater improvement in both depressive symptoms and HTN control than did those in village clinics that provided eCAU.

We are aware of 3 other studies to date that examined collaborative care management approaches for comorbid medical and mental disorders in LMICs in RCTs. Ali and colleagues compared a 12-month, multicomponent collaborative care intervention for depression among adults with type 2 diabetes in urban primary care clinics in India with care as usual [44]. The experimental intervention consisted of components similar to COACH—self-management support and behavioral activation provided by care managers, physicians' decision support tools, and specialist case reviews. They found significantly greater reduction in depressive symptoms and diabetes control than usual care. Post hoc analyses showed greater improvement in systolic BP readings of depressed patients who received the intervention at 12 months, but the improvement was not sustained at 24 months follow-up.

In the second study, Petersen and colleagues employed a collaborative care model for management of comorbid depression and HTN in South African primary care clinics that, in addition to care as usual, included enhanced mental health training for clinic staff and physicians, and clinic-based lay counseling [45]. Care as usual also included decision support for depression care, access to antidepressant prescriptions, and the option of referral to psychologists for treatment. The study found no difference in depression symptom severity or BP readings between groups at 6 months, potentially because so few participants in the experimental intervention received either antidepressant treatment (2.6%) or counselling by lay health workers (7%).

In the third study, Araya and colleagues compared eCAU with a 6-week smartphone-based digital intervention delivering behavioral activation with nurse assistant support to Peruvian and Brazilian adults who had both depression and comorbid HTN or diabetes [46]. Participants who received the digital intervention had significantly greater improvement in

**Table 2. Depression and HTN outcomes in eCAU and COACH groups.**

| | eCAU group (n = 1,133) | COACH group (n = 1,232) | Estimated between-group difference (95% CI) | GLMM analysis | | | Effect size |
| --- | --- | --- | --- | --- | --- | --- | --- |
| | | | | Group effect | Time effect | Group × time interaction | |
| Depressive symptoms HDRS score, mean (SD) | | | | | | | |
| Baseline | 21.76 (3.58) | 22.07 (4.52) | | | | | |
| 3 months | 19.51 (4.87) | 17.97 (5.74) | −2.07 (−3.36, −0.78) | | | | |
| 6 months | 19.58 (4.64) | 15.63 (5.02) | −4.50 −5.79, −3.20 | | | | |
| 9 months | 18.85 (4.61) | 13.77 (4.73) | −5.71 −7.1, −4.41) | | | | |
| 12 months | 18.77 (4.67) | 12.69 (4.22) | −6.67 (−7.97, −5.37) | $F = 52.99, p < 0.001$ | $F = 385.04, p < 0.001$ | $F = 217.38 p < 0.001$ | -1.43[b] (−1.71, −1.15) |
| HTN Controlled[a] –n (%) | | | | | | | |
| Baseline | 229/1,133 (20.21%) | 309/1,232 (25.08%) | | | | | |
| 3 months | 408/1,099 (37.12%) | 665/1,174 (56.64%) | | | | | |
| 6 months | 404/1,078 (37.48%) | 672/1,168 (57.53%) | | | | | |
| 9 months | 435/1,064 (40.88%) | 711/1,141 (62.31%) | | | | | |
| 12 months | 438/1,071 (40.90%) | 827/1,155 (71.60%) | | $F = 34.69 p < 0.001$ | $F = 32.11 p < 0.001$ | $F = 12.74 p < 0.001$ | 18.24[c] (8.40, 39.63) |

Covariates included in the GLMM analysis: religion, employment status, economic satisfaction, WHOQOL-BREF, MOS-SSS-C, Social Network Size, count of comorbidities, HTN control.

COACH, Chinese Older Adults Collaborations in Health; eCAU, enhanced Care-as-Usual; HDRS, Hamilton Depression Rating Scale; HTN, hypertension; GLMM, generalized linear mixed model.

CAU group is the reference group in the model.

[a]HTN Controlled = proportion with controlled HTN per practice guidelines.

[b]Cohen's d (95% confidence interval).

[c]Odds ratio (95% confidence interval).

depressive symptoms at 3 months. However, the impact of the 6-week intervention was not sustained at 6 months. BP readings were not reported.

Although the 3 studies targeted depression in adults with comorbid medical illness in primary care LMIC treatment settings, their collaborative care approaches differed, as did the populations they served and the cultural context in which they were tested. Results were mixed with regard to depression control and largely negative for improvements in BP indices. All 3 included general adult samples, but none focused specifically on older adults, the population with greatest morbidity and mortality associated with comorbid depression and HTN. Additional research is needed to establish the promise of collaborative care management approaches for older adults with these conditions in underresourced primary care settings.

The large effect sizes for both primary outcomes observed with the COACH intervention warrant comment. Depression symptom severity responded robustly to the COACH intervention, significantly greater than in participants who received eCAU. Access to antidepressants and psychiatric consultation may help account for the effect, along with the support of AWs

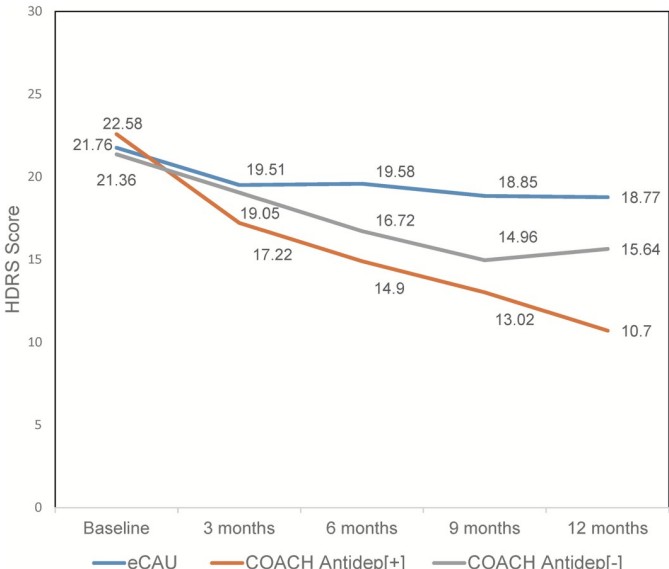

**Fig 2. Depressive symptom severity over 12 months of 3 groups of participants.** eCAU, COACH participants who accepted antidepressant medications (Antidep[+]), and COACH participants who declined antidepressant medications (Antidep[−]). COACH, Chinese Older Adult Collaborations in Health; eCAU, enhanced care-as-usual; HDRS, Hamilton Depression Rating Scale.

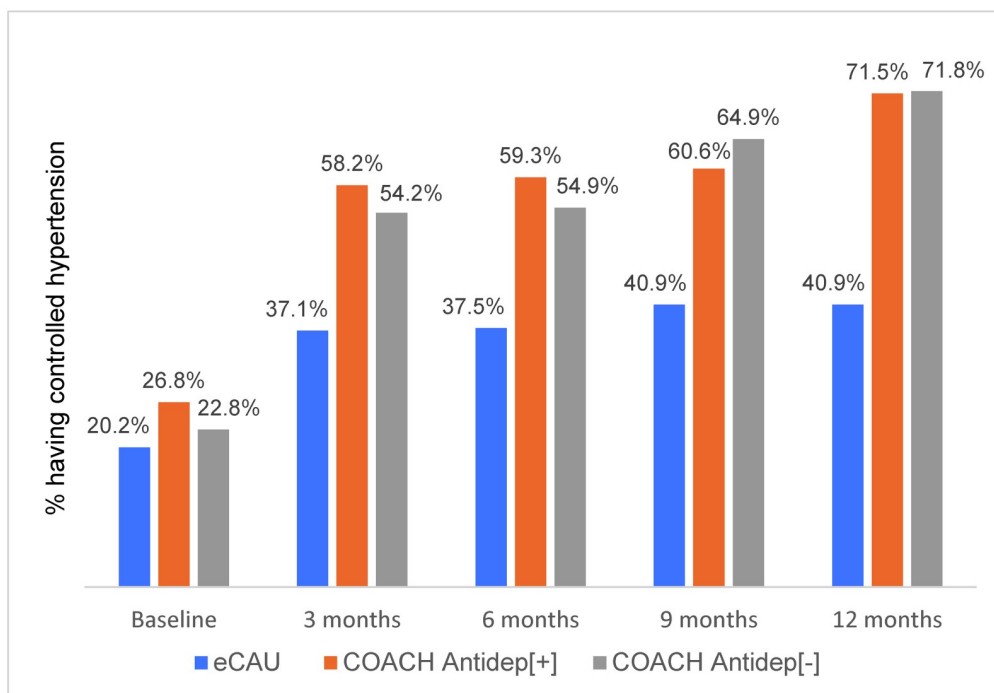

**Fig 3. Proportion of participants in 3 groups who achieved HTN control over 12 months.** eCAU; COACH participants who accepted antidepressant medications (Antidep[+]); COACH participants who declined antidepressant medications (Antidep[−]). COACH, Chinese Older Adult Collaborations in Health; eCAU, enhanced care-as-usual; HTN, hypertension.

with adherence to care and engagement with social supports. Interestingly, the level of improvement in depression outcomes was lower among COACH participants than had been observed in our earlier study in urban primary care clinics [22]. That prior study, which did not examine the impact of the collaborative care intervention on comorbid medical illnesses, included no rural residents and no AW or other lay team member were involved. It did, however, use the same antidepressant medication algorithm as in COACH. In rural clinics delivering the COACH intervention, participants had on average a 43% reduction in HDRS score, whereas older adults who received depression care management using the same antidepressant medication regimen in urban clinics achieved a mean 66% reduction of HDRS total scores over 12 months. Post hoc reasoning suggested that the difference could be accounted for by more uniform antidepressant use in the urban clinic participants (100% received antidepressant treatment versus 58% in COACH). Because the study design did not randomize antidepressant exposure, we could not directly test this hypothesis in the COACH study. However, comparison of the eCAU, Antidep[−], and Antidep[+] groups is suggestive (Fig 2), showing proportional reductions in average HDRS score over 12 months of 14%, 28%, and 53% in the 3 groups, respectively. Further study is needed to establish the priority that should be placed on provision of antidepressants to older adults with clinically significant depressive symptoms in underresourced areas, the great majority of whom will have had no prior exposure to medication treatment. For the substantial minority of COACH participants who are not willing to accept an antidepressant, future research is warranted on incorporation of easily adopted, evidence-based psychosocial interventions like behavioral activation that can be provided by nonspecialists.

The almost 3-fold increase in HTN control rates in COACH intervention recipients could be accounted for by several factors. First, HTN control may have increased with improvement in depressive symptoms due to increases in the hypertensive participant's health behaviors, including activity, diet, and social interactions. The post hoc comparison of those COACH participants who did and did not agree to take an antidepressant showed no difference in the proportions that achieved HTN control, suggesting that the medication was not the driver of improvement. More likely is that the AW, who was charged with supporting participants' adherence to medication and behavioral treatments for HTN, may have positively influenced the outcome as planned. It is also possible, however, that more frequent and more systematic follow-up by AWs and PCPs in the COACH group than in eCAU created more opportunities to detect uncontrolled BP, resulting in more active HTN management. Finally, the older patients and their PCPs were aware that HTN control was a subject of study and may have increased adherence to treatment for that reason. The same explanation may explain the 2-fold increase in HTN control found in the eCAU group as well. Unfortunately, we do not have access to information on rates of adherence to HTN treatment guidelines by participants in either COACH or eCAU groups with which to examine these possibilities further.

Limitations of the study include generalizability of the model to other regions of China or other LMICs [47]. As well, there were methodological weaknesses to which RCTs in rural, underresourced areas are vulnerable. We were unable to mask research assessors to which condition a village was assigned. Although they were kept unaware of the study's hypotheses, there is the risk of assessment bias. Because the study randomized 5 townships to each intervention group while power calculations were made using village as the unit of randomization, the trial could have been underpowered depending on the magnitude of between cluster differences. As anticipated based on the study data, the ICC for the HDRS at the village level (0.064) was greater than at the township level ICC (0.027), well inside our assumed range for the power calculation. Also, there was no meaningful difference between the ICCs for HTN control at the township and village levels (0.025 and 0.021, respectively), and the observed effect sizes were large, indicating that the study is sufficiently powered.

We do not know the extent to which participants followed lifestyle recommendations for management of depression and HTN such as diet, exercise, and socialization. We were unable to assess adherence to medications as planned in the original proposal, precluding examination of the temporal relationships between intervention exposure and changes in depression and HTN control in COACH and eCAU groups.

The integrated care model shown to be effective in primary care settings of high-income countries appears to have promise in rural China for depressed older adults with comorbid medical illness. While the COACH intervention appears effective as administered, however, access to mental healthcare in rural areas remains limited. Further study is needed of modifications to COACH to increase its scalability, such as employment of telehealth technology to link rural village-based COACH teams to psychiatric supports remotely [48], and training of AWs in evidence-based psychotherapies and behavior change techniques to promote a healthy lifestyle [49].

## Supporting information

**S1 Table. Roles and responsibilities of COACH team members.**
(DOCX)

**S2 Table. Systolic and diastolic blood pressure for COACH and eCAU participants at each assessment point.**
(DOCX)

**S3 Table. Expected/observed care management activities by COACH team members.**
(DOCX)

**S4 Table. Depressive symptom severity of study participants over 12 months.**
(DOCX)

**S5 Table. Proportion of study participants with controlled hypertension over 12 months.**
(DOCX)

**S6 Table. Estimated effect size for depression and HTN outcomes in unadjusted analyses.**
(DOCX)

**S1 Appendix. IRB approved protocol, 5-61-13, with annotations of changes made.**
(DOCX)

**S2 Appendix. CONSORT 2010 checklist.**
(DOC)

**S3 Appendix. TIDieR checklist.**
(DOCX)

## Acknowledgments

We thank the rural village residents of Tonglu and Jiande Counties who participated in the study, their primary care doctors, and aging workers for their contributions.

The content is solely the responsibility of the authors and does not necessarily represent the official views of the NIH.

## Author Contributions

**Conceptualization:** Shulin Chen, Yeates Conwell, Lydia Li, Wan Tang, Hillary Bogner, Hengjin Dong.

**Data curation:** Shulin Chen, Yeates Conwell, Jiang Xue, Lydia Li.

**Formal analysis:** Shulin Chen, Jiang Xue, Tingjie Zhao, Wan Tang.

**Funding acquisition:** Shulin Chen, Yeates Conwell.

**Investigation:** Shulin Chen, Yeates Conwell, Jiang Xue.

**Methodology:** Shulin Chen, Yeates Conwell, Jiang Xue, Lydia Li, Hillary Bogner, Hengjin Dong.

**Project administration:** Shulin Chen, Yeates Conwell, Jiang Xue, Lydia Li.

**Resources:** Shulin Chen.

**Supervision:** Shulin Chen, Jiang Xue, Lydia Li.

**Validation:** Shulin Chen, Jiang Xue, Lydia Li, Tingjie Zhao.

**Writing – original draft:** Shulin Chen, Yeates Conwell.

**Writing – review & editing:** Shulin Chen, Yeates Conwell, Jiang Xue, Lydia Li, Tingjie Zhao, Wan Tang, Hillary Bogner, Hengjin Dong.

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
