## [Editor Report · Decision Letter 0]

11 May 2022

Dear Dr CONWELL, 

Thank you for submitting your manuscript entitled "Effectiveness of integrated care for older adults with depression and hypertension in rural China: a cluster randomized controlled trial" for consideration by PLOS Medicine.

Your manuscript has now been evaluated by the PLOS Medicine editorial staff and I am writing to let you know that we would like to send your submission out for external peer review.

Please re-submit your manuscript within two working days, i.e. by May 13 2022 11:59PM.

Kind regards,

Caitlin Moyer, Ph.D.

Associate Editor

PLOS Medicine

---

## [Decision Letter · Decision Letter 1]

28 Jul 2022

Dear Dr. CONWELL,

Thank you very much for submitting your manuscript "Effectiveness of integrated care for older adults with depression and hypertension in rural China: a cluster randomized controlled trial" (PMEDICINE-D-22-01526R1) for consideration at PLOS Medicine. 

Your paper was evaluated by a senior editor and discussed among all the editors here. It was also discussed with an academic editor with relevant expertise, and sent to four independent reviewers, including a statistical reviewer. The reviews are appended at the bottom of this email and any accompanying reviewer attachments can be seen via the link below:

[LINK]

In light of these reviews, I am afraid that we will not be able to accept the manuscript for publication in the journal in its current form, but we would like to consider a revised version that addresses the reviewers' and editors' comments. Obviously we cannot make any decision about publication until we have seen the revised manuscript and your response, and we plan to seek re-review by one or more of the reviewers. 

We expect to receive your revised manuscript by Aug 18 2022 11:59PM. Please email us (plosmedicine@plos.org) if you have any questions or concerns.

We look forward to receiving your revised manuscript. 

Sincerely,

Caitlin Moyer, Ph.D.

Associate Editor

PLOS Medicine

plosmedicine.org

From the Academic Editor:

1. Please provide more comment on the exceptionally large effect size from the HTN treatment arm. There was an approximate three fold increase in control rates in the intervention arm. This is really out of keeping with effect sizes seen in most implementation research trials. It would be helpful if the authors could provide an explanation for this in more detail. It w

2. Please provide a more detailed description of intervention components, using the TIDieR Checklist.

3. Please report on antihypertensive medication use by randomized group. Please also report reduction in blood pressure as a continuous measure. It would be helpful to comment on whether the large effect size over 12 months is related to use of BP medication, or any interventions that were taken to enhance BP medication use.

4.Please report the ICC for primary outcomes (please see CONSORT for Cluster Trials item 17a).

Other editorial points:

5. Data availability statement: Thank you for your willingness to make your data available in the Harvard Dataverse. Please provide the DOI/accession number. PLOS Medicine requires that the de-identified data underlying the specific results in a published article be made available, without restrictions on access, in a public repository or as Supporting Information at the time of article publication, provided it is legal and ethical to do so. Please see the policy at

http://journals.plos.org/plosmedicine/s/data-availability

and FAQs at

http://journals.plos.org/plosmedicine/s/data-availability#loc-faqs-for-data-policy

6. Throughout: Please use “participants” rather than “subjects” when describing the individuals who participated in the trial.

7. Abstract: Please include the trial registration information in the abstract.

8. Abstract: Methods and Findings: Please provide the specific start and end dates of the trial.

9. Abstract: Methods and Findings: Line 41: Please mention how “controlled HTN” was determined.

10. Abstract: Methods and Findings: Please specify who was blinded to the intervention and control.

11. Abstract: Please state that analysis was intention to treat. Please provide the number of participants lost to follow up in each group.

12. Abstract: Methods and Findings: Please quantify the main results with both 95% CIs and p values.

13. Abstract: Methods and Findings: Please provide some summary baseline data/description of the participants.

14. Abstract: Methods and Findings: As mentioned by a reviewer, please focus the abstract on the presentation of primary and secondary pre-specified outcomes, rather than exploratory analyses.

15. Abstract: Methods and Findings: Please include a summary of adverse events if these were assessed in the study.

16. Abstract: Methods and Findings: In the last sentence of the Abstract Methods and Findings section, please describe the main limitation(s) of the study's methodology.

17. Abstract: Conclusions: Please address the study implications without overreaching what can be concluded from the data; the phrase "In this study, we observed ..." may be useful.

18. Author summary: At this stage, we ask that you include a short, non-technical Author Summary of your research to make findings accessible to a wide audience that includes both scientists and non-scientists. The Author Summary should immediately follow the Abstract in your revised manuscript. This text is subject to editorial change and should be distinct from the scientific abstract. Please see our author guidelines for more information: https://journals.plos.org/plosmedicine/s/revising-your-manuscript#loc-author-summary

19. Introduction: Line 69: Here and throughout the manuscript, please refer to high income countries rather than "developed" or "Western" countries.

20. Methods: Protocol: Please include the study protocol document and analysis plan, with any amendments, as Supporting Information to be published with the manuscript if accepted. Please highlight all changes between what was done and what was prespecified in the protocol.

21. Methods: Please ensure that the study is reported according to the CONSORT guideline. Please add the following statement, or similar, to the Methods: "This study is reported as per the Consolidated Standards of Reporting Trials (CONSORT) guideline (S1 Checklist)."

Thank you for including the completed CONSORT checklist as Supporting Information. When completing the checklist, please use section and paragraph numbers (e.g. Methods, paragraph 1), rather than page numbers.

22. Methods: Line 127: Please describe the sociodemographic information collected at baseline. Please describe how social network size was determined.

23. Methods: Line 135: Please provide some brief background on the HDRS measure of depressive symptoms, including scoring.

24. Methods: line 184-185: Please explicitly specify all primary and secondary outcome measures of the study. The following outcomes measures appear to differ between the submitted manuscript and the trial registry: The registered primary outcome is Depressive Symptom Change and HTN control is not mentioned, except as an Other Outcome Measure. Health related quality of life at each timepoint is not described. Costs associated with intervention are not described. Assessment of outcomes at time points prior to the 12 month assessment are also not described. Please clarify and explain the discrepancy. If the outcomes were not prespecified in the protocol, please indicate that they were post hoc and explain why they were added. Post hoc comparisons should be presented as hypothesis generating rather than conclusive.

25. Methods: Line 243-244: Please define "lost to follow-up" as used in this study. Other reasons for exclusion should be defined.

26. Results: Please present numerators and denominators for percentages, at least in the Tables [not necessarily each time they're mentioned].

27. Results: Line 255-263: Please quantify results presented in the text, 95% CIs and p values for depression scores and HTN outcomes, as well as group x time interactions.

28. Results: Line 266-275: Please quantify the comparison between those who did and did not accept prescribed antidepressants, in terms of differences in HTN control and HDRS scores.

29. Discussion: Please present and organize the Discussion as follows: a short, clear summary of the article's findings; what the study adds to existing research and where and why the results may differ from previous research; strengths and limitations of the study; implications and next steps for research, clinical practice, and/or public policy; one-paragraph conclusion.

30. Figure 2 and Figure 3: Please also present these data in Table format.

31. Figure 2: Please show the axis beginning at zero. If this is not possible, please show a break in the axis.

Comments from the reviewers:

Reviewer #1: Statistical review

This paper reports a cluster randomised trial evaluating an intervention aimed at improving mental health and hypertension in over 60s in Chinese rural settings. The intervention significant improved both outcomes. 

The trial is well reported and used appropriate statistical methods. I only had some minor comments to make.

1. Abstract, methods and findings "study was a 12-month cluster randomized controlled" - I would move the 12-month part to the end of the sentence and reword as 'with 12 month follow-up'. Currently it reads strangely that the trial was 12 months but took place between 2014 and 2017.

2. Abstract line 43: I would add p-values to the estimated effect sizes and CIs, as the trial was prospectively powered to test hypotheses. I do not think it is good practice, according to CONSORT, to report non-pre-specified analyses in the abstract (it's fine to report in results), but will leave that to the editor.

3. Discussion: was there a reason for strong differences in baseline between randomised groups? Presumably this is just due to the small number of clusters? Generally it is not advised to test for difference in baseline variables in RCTs but I would imagine the reasons for this would not apply to CRTs with small number of clusters.

James Wason

Reviewer #2: This is a welcome paper in the realm of physical and mental health multimorbidity from an LMIC and shows gratifying improvements in blood pressure control with the multifaceted COACH intervention. 

There are a few points that need clarification

ABSTRACT

1: It is not clear how villages can be subjects as it stated

2 Subjects were rural village clinics of randomly selected towns in Zhejiang Province, China

Please clarify how villages can be part of towns- this should be added to the methods in order to provide greater context for the reader

If the study was a 12-month cluster randomized controlled trial conducted from 2014 through 2017, when were the follow up assessments made and why were the analyses only conducted in 2020-2021

METHODS:

How were the participants identified - was it on the day of attendance or from registries?

It is not clear what the frequency of visits in the enhanced usual care was- was it the same as for the intervention group?

How generalisable would it be for psychiatrists to be available for primary care clinic visits in rural China outside cities?

What is the explanation for the marked increase in blood pressure control in the EAU group

It is unfortunate that there were so many baseline differences between the 2 groups but the appropriate analyses were undertaken to account for these

DISCUSSION

In the discussion two other RCTs are referred to that have examined collaborative care management of comorbid medical and mental disorders in LMIC- the one for depression and diabetes and the other depression and hypertension. The first demonstrated greater improvement in depression and DM control but no benefit was found in the latter. It would be helpful for the authors to tease out the components of the collaborative care in these three trials in an attempt to better understand why there were these differences. For example what role did the psychiatrists play in the current trial- one would assume a major one given the lack of difference in outcome in people who were on antidepressants compared to those who did not accept them. Some more detail of how the AWs supported the intervention group would be helpful. 

Reviewer #3: The authors should be congratulated for conducting and reporting on a remarkable trial of an integrated intervention for late-life depression and hypertension in rural China. This is the single largest study of its kind and it has multiple strengths including the large sample size (both in terms of the number of clinics and the number of patients enrolled) and also the high rate of enrollment and follow-up among eligible subjects. The intervention and the training efforts seem 'realistic' and should not be too challenging to replicate in other 'real world' settings. I do have a few suggestions for the final version of this manuscript. 

It is unfortunate that it was not possible to blind the assessors to intervention status. This could create rater biases that are hard to control for, especially since one of the main outcome measures (HDRS) is not a patient rating but an assessor-rated scale. 

The relative effectiveness of the intervention seems to be much greater for hypertension control than for depression. Can the authors comment on this in the discussion? The base rate of controlled hypertension was remarkably low and the effectiveness of the intervention on achieving control of hypertension was remarkably high - so from a public health perspective, this seems to be the primary benefit of this intervention. It is not clear from the paper if and how the focus on depression helped with hypertension control or what was done specifically in the intervention to achieve hypertension control. It is also not clear if a similar algorithm driven intervention focusing on blood pressure control could have achieved similar results for hypertension control. It would be helpful to have more information on changes and differences in the use of antihypertensive medications. It may also be possible that more frequent and more systematic follow-up simply created more opportunities for PCPs and their staff to detect the fact that hypertension was not (yet) controlled, resulting in more active hypertension management.

The effects on depression are significant but moderate. Several things may contribute to this. First, the benefits of antidepressant medications may be relatively greater for more severe forms of depression - and given that subjects were included who had a PHQ-9 screen of 10 or greater, there may be a number of patients with milder forms of depression who did not have much relative benefit from medications. Secondly, as the authors point out, nearly half of participants did not agree to take antidepressants. It might be helpful to augment the intervention with some relatively simple psychosocial intervention such as behavioral activation strategies that can be effective alternatives to medication or augmentation strategies and that can be provided by non-specialists. 

Could the authors comment on baseline differences in depression severity and the WHOQOL-BREF. What might contribute to these differences and how where they addressed? 

One minor comment: Page 10, line 208 - we cannot say for certain that there was a change in 'depressive disorders' but rather what was observed was a change in the severity of depression symptoms as assessed by the HDRS. 

Reviewer #4: This is a relevant article targeting rural village clinics in Zhejiang Province, China, to deliver the Chinese Older Adult Collaborations in Health (COACH) intervention or enhanced care-as-usual (eCAU). The COACH intervention consisted of algorithm-driven treatment of depression and HTN by village primary care doctors supported by village lay workers with consultation from centrally-located psychiatrists. The primary outcomes of the study were change in depression symptom severity as measured by the HDRS at baseline and follow-up assessments, and the proportions (%) of subjects whose HTN was controlled. 

Major comments

1. Introduction, lines 71-73, refers to a previous study in China by the same authors (PMID: 26360086). Could you expand what is the difference with the current study and why this study was needed?

2. Introduction, lines 71-72, the statement "Rarely, however, have studies been conducted in LMICs that used a randomized control trial (RCT) design" misses the most recent evidence arising from Latin America (see PMID: 33974019), which could complement this study. Again, this reference will also be uniquely suited to enhance the discussion provided in lines 290-297.

3. Statistical analysis, lines 190-201, the authors present a detectable between-group proportion (for HTN control) indicating that calculated values of 11% to 17% are "well within the range of clinically meaningful differences in primary care settings". Yet, we this reviewer would like to see a similar remark for the depression outcomes, since the authors only report that "detectable effect size ranged from 0.17 to 0.26 for the continuous outcome of depression symptom change" and the reader does not know if such calculations imply clinically meaningful differences.

4. Methods, an ethical procedures section is missing. The information is scattered in other sections.

5. Table 1 does not require p-values, as p-value is a measure for inferential purposes, not descriptive ones (PMID: 30096356).

6. Results lacks a description of the delivery of the intervention, how many aging workers (AW), what was the turnover, how many replacements were needed, was re-training needed, was village-level or town-level tailoring required, how often did consultations with psychiatrists occur, etc., etc. This is a complex intervention that requires aligning to the TIDieR checklist (PMID: 24609605)

Minor comments

7. Abstract, lines 31-32, please clarify "Subjects were rural village clinics of randomly selected towns in Zhejiang Province, China."

8. Line 237 reads "The study protocol was prospectively registered with Grants.gov [41]", yet the correct Website is https://clinicaltrials.gov/

Discretionary comments

9. Abstract, line 53: "geriatric mental health care is lacking", do you mean minimal or absent?

[LINK]

---

## [Decision Letter · Decision Letter 2]

27 Sep 2022

Dear Dr. CONWELL,

Thank you very much for re-submitting your manuscript "Effectiveness of integrated care for older adults with depression and hypertension in rural China: a cluster randomized controlled trial" (PMEDICINE-D-22-01526R2) for review by PLOS Medicine.

I have discussed the paper with my colleagues and the academic editor and it was also seen again by three reviewers. I am pleased to say that provided the remaining editorial and production issues are dealt with we are planning to accept the paper for publication in the journal.

[LINK]

We look forward to receiving the revised manuscript by Oct 04 2022 11:59PM.   

Sincerely,

Callam Davidson, 

Associate Editor 

PLOS Medicine

plosmedicine.org

Requests from Editors:

Lines 59-60: Please do not report P=0.0001; report as P < 0.001.

Abstract: Please define the following abbreviations on first use: HTN, PHQ, PCP, OR, LMIC (abbreviations are only necessary if the term is used more than once).

Author Summary: Please include the headline numbers from the study, such as the sample size and key findings.

Please update your TIDieR checklist to use section and paragraph numbers rather than page numbers.

Line 342: Please change text colour to black.

Table 2: Please consider presenting the crude (unadjusted) analysis as Supporting Information.

Line 393: S1 Table does not contain information regarding demographic and clinical characteristics at baseline, please check whether this citation is incorrect. 

Line 526: Please remove ‘however’.

Comments from Reviewers:

Reviewer #1: Thank you to the authors for addressing my previous comments well. I have no further issues to raise.

Reviewer #3: I am satisfied with the response to the initial reviews and suggest publication without additional need for modifications.

Reviewer #4: The current version is a much more improved version of the manuscript and I am satisfied with the changes made. I am also grateful to the authors for providing a detailed description of the intervention and using TIDieR as a guidance, which will improve towards the reproducibility of the research but also will guide the reader on the extent of activities required to be in place to consider a similar approach.

I have only one two comments left

1. Being a trial, I think you need a subheading in the methods describing the main outcomes

2. Related to that point, I would encourage the authors/editors to maintain uniformity in the entire document, including text, table and figures about the sequence of how the outcomes are presented (e.g. depression goes always first and hypertension second).

Thank you for the opportunity to see this version.

[LINK]

---

## [Editor Report · Decision Letter 3]

4 Oct 2022

Dear Dr CONWELL, 

On behalf of my colleagues and the Academic Editor, Professor David Peiris, I am pleased to inform you that we have agreed to publish your manuscript "Effectiveness of integrated care for older adults with depression and hypertension in rural China: a cluster randomized controlled trial" (PMEDICINE-D-22-01526R3) in PLOS Medicine.

PRESS

Sincerely, 

Callam Davidson 

Associate Editor 

PLOS Medicine